# The importance and availability of adjustments to improve access for autistic adults who need mental and physical healthcare: findings from UK surveys

Samuel Brice  ,[1] Jacqui Rodgers,[1] Barry Ingham,[1,2] David Mason,[1] Colin Wilson,[1] Mark Freeston,[3] Ann Le Couteur,[1] Jeremy R Parr[1,2]

¹Population Health Sciences Institute, Newcastle University, Newcastle upon Tyne, UK
²Cumbria, Northumberland, Tyne and Wear NHS Foundation Trust, Newcastle upon Tyne, UK
³School of Psychology, Newcastle University, Newcastle upon Tyne, UK

**Correspondence to**
Professor Jeremy R Parr;
Jeremy.Parr@ncl.ac.uk

## ABSTRACT

**Objectives** To investigate autistic people's views on the importance and availability of adjustments to mental and physical healthcare provision. To explore whether specific categories of adjustments can be identified and to identify any differences in their importance and availability between mental and physical healthcare.

**Design** Data from two studies, both employing a cross-sectional survey design.

**Setting** UK-based autistic adults registered with the Adult Autism Spectrum Cohort-UK were contacted by post or online. In both studies, recruitment was staged over a 12-month period. Non-responders were sent a single reminder letter 2 weeks after initial contact.

**Participants** 537 autistic adults completed a survey about mental health services (51% response rate), 407 completed the physical health survey (49% response rate). Within these samples, 221 participants completed both surveys.

**Primary outcome measures** Each study developed a bespoke survey to explore participants' views on mental and physical health services, respectively. Both included an identical list of adjustments that participants rated based on importance and availability.

**Results** Three factors of important adjustments were identified: sensory environment, clinical and service context, and clinician knowledge and communication. Adjustments across healthcare settings were widely rated as being important yet rarely available. One significant difference between the importance of adjustments available through mental and physical health services was identified. Participants reported that having access to a clinician who is willing to adapt their approach to suit the person's preferences was significantly more important for participants attending mental health settings (p=0.001).

**Conclusions** Autistic people reported the limited availability of important adjustments in current healthcare provision. To address unmet need and tackle the health inequalities faced by autistic people attending physical and mental healthcare settings, healthcare providers should offer adjustments relating to the three identified factors. Future research should focus on identifying and addressing service provider barriers to implementation.

## Strengths and limitations of this study

► This research matched published research priorities of the autism community and focused directly on the experiences of autistic people.
► This study reports on key adjustments to both physical and mental health services.
► Our data-driven approach enabled us to determine overarching categories of adjustments that could be used to inform both policy and practice in relation to service development.
► A notable proportion of participants reported that they did not have sufficient knowledge to comment on the availability of adjustments.
► This study did not seek the opinions of health service providers, which should be a focus for future implementation research.

## INTRODUCTION

Autism spectrum disorder[i] is characterised by lifelong persistent difficulties with social communication and social interaction and the presence of patterns of restrictive and repetitive behaviours or interests that may include hyper-reactivity or hyporeactivity to sensory aspects of the environment.[1] At least 1.1% of the UK population is autistic.[2]

Autistic people are more likely to experience mental and physical health conditions than the general (neurotypical) population[3]; this persists across the lifespan.[4] Poor health has been shown to predict poorer quality of life.[5 6] Autistic adults are more

---

[i]We will use the term 'autism' or 'autistic person' throughout this paper, reflecting a growing consensus that thisterminology is preferred by the autism community. This terminology includes previously recognised subcategories of autism, including but not limited to Asperger's Disorder, Autistic Disorder and Pervasive Developmental Disorder.

likely than neurotypical adults to experience physical health conditions, such as epilepsy, cardiovascular disease and diabetes.[7] Anxiety and depression are particularly common, with an estimated lifetime prevalence of around 40%.[8 9] Autistic people are also more likely to experience multiple mental and physical health conditions concurrently.[10] Epidemiological studies suggest autistic people are also significantly more likely to face premature mortality: the average life expectancy for an autistic adult without intellectual disability is 12 years lower than for neurotypical people.[11]

## Barriers to healthcare access

Autistic adults face a number of significant barriers to effective healthcare access and treatment.[12 13] To date, the most frequently reported barriers have been considered within three factors: individual-level (eg, barriers to effective communication with clinicians, sensory sensitivities), provider level (eg, lack of 'autism awareness' among clinicians) and system level (eg, convoluted referral pathways and lack of availability of specialist services). Although such attempts to categorise adjustments in this way are useful to provide a simplified message to healthcare providers, this categorisation system has yet to receive empirical support. Recent research indicates that these barriers prevent autistic people from arranging and attending healthcare appointments.[14] In addition, other individual factors, such as difficulties coping with the potential uncertainty of healthcare situations[15] and pre-existing anxiety,[8] may make the experience of attending healthcare appointments especially problematic for autistic people. Further, although the UK National Health Service (NHS) is publicly funded and free at the point of access, in other countries the affordability of healthcare has been shown to be a barrier that is disproportionately more likely to affect autistic people.[16] There is also evidence of a lack of training, skills and/or confidence among healthcare professionals in working with and delivering interventions to autistic people,[17] which may impact on their continued engagement with healthcare services.[12]

UK autism research priorities focus on aspects of physical and mental healthcare, including 'How should service delivery for autistic people be improved and adapted in order to meet their needs?'.[18] Many autistic people report dissatisfaction with the current accessibility of healthcare across both physical and mental health settings.[19 20] Several recent UK clinical guidelines have recommended that to reduce barriers to healthcare, providers should adjust their procedures and approach to best meet the individual needs of autistic people, with additional specific guidance for mental health services, to meet the needs of people with intellectual disabilities.[21–23] A recent study concluded that 'adjustments for communication needs are as necessary for autistic people as ramps are for wheelchair users'.[14] Evidence suggests that adjustments to services such as reducing exposure to potentially aversive sensory stimuli (eg, noisy waiting rooms) and the

clinician using the autistic adult's preferred communication methods, do facilitate effective healthcare access for autistic people.[13] However, the importance and availability of specific adjustments across healthcare settings is unknown.

## Aims

1. To investigate autistic people's views on the importance and availability of adjustments that are designed to promote the accessibility and acceptability of mental and physical healthcare provision.
2. To explore whether specific categories of adjustments can be identified using exploratory factor analysis (EFA) and to identify any differences in their importance and availability between mental and physical healthcare.

## METHOD

### Design

This study reports on data from two UK studies—one focused on anxiety (as an exemplar of a mental health condition) and the other focused on physical health conditions. Both employed a cross-sectional survey design and were undertaken in 2018–2019.

### Participants

Participants for both studies were recruited via the Adult Autism Spectrum Cohort-UK (ASC-UK; https://research.ncl.ac.uk/adultautismspectrum/), a longitudinal cohort study of autistic adults. ASC-UK recruits from a wide variety of UK sources including NHS diagnostic/health services, charitable/community organisations and also receives self-referrals. In line with previous research[6], these studies included some participants who were awaiting autism assessment or suspected that they were autistic (18% and 16% of mental health and physical health samples, respectively).

Participants completed paper or web-based materials, depending on their preference. Following consent, participants completed a registration questionnaire that included questions on demographics, mental and physical health, education, employment and aspects of daily living. Participants also completed the Social Responsiveness Scale-2nd Edition (SRS-2[24]).

The Personalised Anxiety Treatment-Autism (PAT-A) study investigated autistic people's experiences of anxiety and access to mental healthcare, with a particular focus on accessing psychological therapies, the primary treatment choice for anxiety conditions in UK clinical guidance.[22] Inclusion criteria were autistic people living in the UK aged 18 years and older. A total of 1113 autistic people aged 18 and over who had previously reported a diagnosed or suspected anxiety condition were invited to participate. A total of 568 (51%) returned the consent form and completed the survey. Thirty-one participants consented and did not respond to any of the questions about adjustments and were not included in analyses, resulting in a final sample of 537.

The Improving the Health of Autistic People (IHOAP) study investigated the physical health of autistic people, and experiences of accessing healthcare for physical health conditions, including impact on daily life, screening services offered and taken up, and factors that made accessing healthcare difficult. Inclusion criteria were autistic people living in the UK aged 18 years and older. A total of 945 autistic adults were invited to participate; 461 (49%) people consented and completed the survey. Fifty-four participants did not respond to any survey questions about adjustments and were, therefore, not included in analyses, resulting in a final sample of 407.

There were 723 unique participants in the two studies, including 221 participants who completed both studies. Demographic information (from ASC-UK) for participants from the PAT-A and IHOAP studies is provided in table 1.

## Procedure

Eligible participants were sent an information sheet (including an 'easyread' version), consent form and survey, either by post or email depending on their contact preferences. Those preferring to be contacted online were sent a unique link based on a participant number and completed the consent form and survey using Qualtrics.[25] Participants who received materials by post completed a written consent form and survey and returned this to the research team in a prepaid envelope. Written informed consent was provided by all participants. Participants who had not responded after 2 weeks were sent a single reminder letter. Recruitment was staged over a 12-month period. Participation was not incentivised.

## Patient and public involvement

This research is directly matched to published research priorities of the autism community.[18] The ASC-UK cohort study involved autistic people at the design stage as well as in the design and delivery of ongoing associated projects. Prior to commencing both studies, patient and public involvement focus groups were convened with community stakeholders to discuss and agree to the methods and materials, including survey questions. An autistic person and parent of an autistic person are coinvestigators on this research. Author CW is autistic and was involved with the analysis of data and preparation of this manuscript. Our dissemination plan includes consideration of the most effective ways to share these findings with autistic people, including presentations and newsletter updates.

## Measures

Both samples completed the same survey about adjustments to services. Both samples also provided data on prior service use and experiences of being offered adjustments. Other data were also collected as part of the PAT-A and IHOAP studies and will be reported separately (manuscripts in process). The survey questionnaires included a list of adjustments previously reported to improve access, inclusivity and appropriateness of health services, together with an option for open text responses to record any other adjustments. The list was based on published evidence,[13] clinical experience and the Reasonable Adjustments Flag (NHS Digital, https://digital.nhs.uk/services/reasonable-adjustment-flag[26]). The list of adjustments can be seen in tables 2 and 3.

Participants were asked to rate the importance of each adjustment on a five-point Likert scale from (1) not at all important; (2) not very important; (3) neither important nor unimportant; (4) somewhat important and (5) very important. Participants also rated how available they thought each adjustment was using a five-point scale (1) never available, (2) rarely available, (3) available about half the time, (4) available most of the time, (5) available all of the time. For availability, a sixth option 'I don't know' was provided to enable those who did not feel that they had enough experience of that particular adjustment to report this, enabling the research team to discriminate between lack of experience and a missing response. Any additional adjustments identified by participants in open text response boxes were recorded.

The SRS-2[24]; is a widely used standardised self-report questionnaire used to characterise autism 'severity'. The SRS-2 has demonstrated excellent internal consistency and the predictive validity of the adult form has been reported to have a specificity level of 0.60 and a sensitivity of 0.86.[27] The full-scale score was used to make comparisons between those who self-reported a diagnosis of autism and those who suspected they had autism or were awaiting assessment; and between responders and non-responders (further detail in table 1, online supplemental table 1).

## Analysis

Key demographic characteristics (age, gender and level of autism traits) were compared between groups using independent t-tests to compare means or $\chi^2$ tests to compare the distribution of categorical variables. Comparisons were made between participants who self-reported an autism diagnosis, and those who suspected that they were autistic or were awaiting an autism diagnosis; responders versus non-responders (in both the mental health and physical health studies) and between participants from the mental health and physical health samples. These comparisons were made to assess the representativeness and comparability of the samples and to aid with interpretation of the findings. Participants who completed both studies were not included in any statistical comparisons made between the mental health and physical health samples to avoid double counting and so breaking the assumption of independence.

EFA was conducted in order to assess whether the adjustments clustered together in a meaningful way. FACTOR software V.10.8.04[28] was used to conduct EFA on data on the importance of each adjustment in mental health and physical health services separately. Polychoric correlations were used as this is deemed most appropriate for use with ordinal and/or skewed data such as these.[29] Unweighted

**Table 1** Comparison of the demographic characteristics of participants included in the mental health and physical health samples

| Variable | Mental health sample (PAT-A study) | Physical health sample (IHOAP study) |
|---|---|---|
| **Total N** | 537 | 407 |
| **Gender N (%)** | | |
| Male | 234 (43.6) | 167 (41.0) |
| Female | 281 (52.3) | 227 (55.8) |
| Other/rather not say | 22 (4.1) | 13 (3.2) |
| **Age: mean (SD) (range)** | 41.3 (13.8) (18.0–77.0) | 44.4 (13.4) (18.0–79.0) |
| **Ethnicity N (%)** | | |
| White British | 501 (93.3) | 374 (91.9) |
| Asian | 3 (0.6) | 1 (0.2) |
| Black | 2 (0.4) | 3 (0.7) |
| Mixed | 10 (1.9) | 8 (2.0) |
| Other/rather not say | 8 (1.5) | 7 (1.7) |
| Not reported | 13 (2.4) | 14 (3.4) |
| **Highest education N (%)** | | |
| Postgraduate degree | 94 (17.5) | 72 (17.7) |
| Bachelor's degree | 135 (25.1) | 97 (23.8) |
| Diploma of higher education | 34 (6.3) | 28 (6.9) |
| Certificate of higher education | 18 (3.4) | 17 (4.2) |
| A-level | 88 (16.4) | 76 (18.7) |
| General Certificate of Secondary Education (GCSE) | 108 (20.1) | 71 (17.4) |
| Basic skills | 20 (3.7) | 10 (2.5) |
| No formal qualifications | 32 (6.0) | 30 (7.4) |
| Other | 8 (1.5) | 6 (1.5) |
| **Employment status N (%)** | | |
| Employed without support | 213 (39.7) | 154 (37.8) |
| Employed with support | 9 (1.7) | 5 (1.2) |
| Volunteer | 49 (9.1) | 38 (9.3) |
| Unemployed | 170 (31.7) | 137 (33.7) |
| Retired | 30 (5.6) | 25 (6.1) |
| Other | 58 (10.8) | 38 (9.3) |
| No response | 8 (1.5) | 10 (2.5) |
| **ASD diagnosis N (%)** | | |
| Formal diagnosis | 443 (82.5) | 341 (83.8) |
| Suspected/unsure/awaiting assessment | 94 (17.5) | 66 (16.2) |
| Age at diagnosis: mean (SD) (range) | 35.8 (15.8) (3.0–73.0) | 38.7 (14.8) (2.0–68.0) |
| **SRS score† by severity category N (%)** | | |
| Normal | 20 (4.2) | 9 (2.5) |
| Mild | 45 (9.4) | 35 (9.7) |
| Moderate | 172 (35.8) | 115 (31.9) |
| Severe | 244 (50.7) | 201 (55.8) |
| Mean (SD) | 110.5 (25.6) | 115.1 (25.8) |
| **Support received N (%)‡** | | |
| Home | 152 (15.5) | 115 (15.2) |

Continued

| Table 1 | Continued | | |
| --- | --- | --- | --- |
| **Variable** | | **Mental health sample (PAT-A study)** | **Physical health sample (IHOAP study)** |
| Employment | | 50 (5.1) | 36 (4.8) |
| Health | | 110 (11.2) | 95 (12.6) |
| Finance | | 129 (13.2) | 99 (13.1) |
| Social | | 75 (7.7) | 57 (7.5) |
| Lifelong learning | | 67 (6.9) | 47 (6.2) |
| Community | | 78 (8.0) | 68 (9.0) |
| Organisation | | 73 (7.5) | 55 (7.3) |
| Do not receive support | | 244 (24.9) | 183 (24.2) |
| **Mental health condition N (%)** | | | |
| Diagnosis | | – | 351 (86.2) |
| No diagnosis or suspected | | – | 56 (13.8) |
| **Physical health condition N (%)** | | | |
| Diagnosis | | 397 (73.9) | – |
| No diagnosis or suspected | | 140 (26.1) | – |
| **Prior service use (anxiety) N (%)** | | | |
| Prescribed medication | | 394 (73.4) | – |
| Referred for psychological therapy | | 400 (76.5) | – |
| Completed treatment | | 320 (80.0) | – |
| Did not complete/attend | | 80 (20.0) | – |
| Offered adjustments (mental health services) | | 90 (24.3) | – |
| **Prior service use (physical health) N (%)** | | | |
| Seen GP in the preceding 12 months | | – | 375 (92.0) |
| Seen specialist in the preceding 24 months | | – | 240 (59.0) |
| Require regular appointments due to a medical condition | | – | 240 (59.0) |
| Missed at least one appointment with a specialist | | – | 58 (24.2) |
| Offered adjustments (physical health services) | | – | 75 (19.8) |

*Includes self-employed.
†Not available for all participants.
‡May add to more than total sample size as multiple responses were allowed.
ASD, autism spectrum disorder; GP, general practitioner; IHOAP, Improving the Health of Autistic People; PAT-A, Personalised Anxiety Treatment-Autism; SRS, Social Responsiveness Scale–second edition.

least squares were used to extract the factors and the rotation method was Promin. The use of an oblique method of rotation such as this allows for the correlation between factors to be calculated, which is in line with best scientific practice.[30] Bias correlated bootstrap resampling was undertaken using 5000 bootstrap samples. This decision was taken in order to reduce the influence of any outliers and maximise the generalisability of the sample. Model fit was assessed using the following indices: Root Mean Square Error of Approximation, Non-Normed Fit Index, Comparative Fit Index, Goodness of Fit Index and root mean square of residuals.

Participants with at least one missing response on any of the importance of adjustment ratings were deleted listwise prior to conducting EFA as any efforts to impute missing items may have biased the results at this stage.

This resulted in the removal of 51 participants from the mental health dataset (final N=486 and 78 participants from the physical health dataset (N=329). Sample size suitability for EFA has been a matter of debate, however best practice guidelines suggest that a participant-to-item ratio of between 5:1 and 10:1 is considered to be suitable.[31] Therefore, the 27:1 and 18.3/1 participant-to-item ratios in the mental health and physical health datasets, respectively, were deemed more than adequate for EFA. Furthermore, the Kaiser-Meyer-Olkin (KMO) test revealed that the sampling adequacy was 'very good' in both datasets (mental health: KMO=0.92; physical health: KMO=0.91).

Decisions regarding the number of factors to retain was based on best practice guidelines,[31] which included consulting existing theoretical knowledge, examining the

scree plots, retaining factors with an eigenvalue greater than one and conducting parallel analysis. These indicators supported the retention of between two and four factors in each of the datasets. An a priori loading criterion of ≥0.4 was applied to the rotated loading matrices to ensure that emergent factors were meaningful, while not being so stringent as to limit the reproducibility of the findings.[32]

Quantitative data were analysed descriptively. Analysis of open text comments and descriptions of experiences about adjustments used framework analysis.[33] Two researchers (JR and SB) coded open-text responses against the adjustment categories indicated in EFA and any discrepancies were discussed and agreed on. Additional adjustments that did not fit within these categories were recorded under the category 'other'.

Service-level differences (ie, comparisons between mental healthcare and physical healthcare) in adjustments that were reported by autistic people as most important and least available were explored using McNemar's test to compare nominal data in the paired sample of 221 participants who completed the PAT-A and IHOAP surveys. Variables were first dichotomised to separate the proportion who selected that an adjustment was 'very important' compared with all other levels of importance and 'never available' compared with all other levels of availability. McNemar's comparisons were computed for each of the 18 core adjustments based on the dichotomised importance and availability ratings. Simes' modified Bonferroni correction[34] was used to minimise risk of type I error.

## RESULTS

Details of statistical comparisons between the study participants and those who did not respond, and within and between subsamples (ie, mental health and physical health samples) can be seen in online supplemental table 1. There was no significant difference in autism characteristics (SRS-2 total score) between those who reported an autism spectrum diagnosis and those who suspected they were autistic or were awaiting diagnosis in all comparison groups. The only significant differences were: the physical health sample (mean age=44.4, SD=13.4) were significantly older than the mental health sample (mean age=41.3, SD=13.8); those who took part in the physical health study were significantly older (mean age=44.4, SD=13.4) than non-responders (mean age=41.0, SD=13.3).

### Adjustments and service use

Sixty-nine per cent (n=193) of autistic people who were not offered adjustments by mental health services and 56% (n=171) of those not offered adjustments by physical health services suggested that adjustments would have been required or helpful. Of those who attended at least one session of psychological therapy but did not complete their treatment 64% (n=40) cited a lack of adjustments to

meet their needs as a contributing factor. Poor accessibility or lack of adjustments were a factor for 45% (n=26) of those who reported missing at least one physical health appointment.

### Categorising key adjustments to services

EFA supported the retention of between two and four factors. Using an a priori loading criterion of 0.4 a clear three-factor solution was retained in the mental health data (factor loadings of >0.4 for 17/18 items; intercorrelations: 0.69 to 0.79). The factors related to adjustments for (1) 'sensory environment', (2) 'clinical and service context' and (3) 'clinician knowledge and communication'. The adjustment 'short waiting times to be seen when you attend appointments' did not load clearly onto any factor. This solution demonstrated very good/excellent model fit across all tested indices. A four-factor solution was most interpretable for the physical health data (factor loadings of >0.4 for 15/18 items and good model fit). Closer inspection indicated strong similarities across both solutions. Thirteen out of 18 adjustments loaded onto the same, or similar factors. Data for the item relating to shorter waiting times are presented alone (see online supplemental tables 2–4) for factor loadings and model fit indices. The importance (table 2) and availability (table 3) of adjustments are presented in the categories identified by factor analysis. Full response frequencies for the importance of adjustments can be seen in online supplemental table 5 and availability of adjustments in online supplemental table 6.

Most respondents reported that all adjustments were 'somewhat important' or 'very important' (hereafter referred to as important) in both mental and physical health services. In mental health services, those rating each adjustment as important (ie, 'somewhat important' or 'very important') ranged from 46% (n=240) (access to online appointments) to 98% (n=524) (clinician who understands autism). In physical health services, these ranged from 47% (n=178) (changing frequency of appointments) to 97% (n=386) (clinician who understands autism). High proportions of participants also rated each adjustment as 'very important' ranging from 23% (n=120) (access to online appointments) to 88% (n=468) (clinician who understands autism) in mental health services and 18% (n=69) (changing frequency of appointments) to 81% (n=322) (clinician who understands autism) in physical health services. Autistic adults rated 'having access to a clinician who understands autism' as the most important adjustment in both mental health (98.3%, n=524) and physical health (96.5%, n=386) settings.

For over half of respondents all but three adjustments were endorsed as 'unavailable' (ie, rarely available' or 'never available') across both health service contexts. Adjustments were frequently endorsed as being 'never available' across both settings, ranging from 10% (n=31) to 63% (n=145) in mental health and 13% (n=38) to 67% (n=138) in physical health. In both cases, the

**Table 2** The importance of key adjustments to mental and physical health services to meet the needs of autistic adults

| Key adjustment | Mental health services | | Physical health services | |
|---|---|---|---|---|
| | Somewhat important % (N) | Very important % (N) | Somewhat important % (N) | Very important % (N) |
| **Sensory environment** | | | | |
| Change the sensory environment in the building that the appointment will take place in | 33.6 (178) | 39.4 (209) | 31.5 (123) | 42.5 (166) |
| Locations (eg, waiting rooms) with small numbers of people | 30.6 (162) | 50.7 (268) | 29.4 (116) | 52.8 (208) |
| Locations with low noise levels | 28.2 (149) | 60.9 (322) | 26.4 (104) | 61.9 (244) |
| Locations with low light levels | 29.1 (153) | 35.2 (185) | 31.3 (121) | 35.7 (138) |
| **Clinical and service context** | | | | |
| Changing the length of appointments to suit you | 33.8 (181) | 36.2 (194) | 33.9 (134) | 44.8 (177) |
| Offering appointments online or via apps | 22.9 (120) | 22.9 (120) | 18.8 (73) | 31.2 (121) |
| Changing how often you are asked to attend appointments | 30.7 (163) | 24.5 (130) | 28.6 (109) | 18.1 (69) |
| Give information to the clinician pre-appointment so that they can prepare | 27.7 (147) | 60.6 (322) | 32.8 (130) | 48.0 (190) |
| Provide support in relation to attending appointments (eg, managing fears or uncertainties which might make attending difficult) | 27.2 (146) | 53.5 (282) | 28.0 (109) | 50.9 (198) |
| Appropriate distractions provided while waiting to be seen at appointment (eg, tablet with headphones) | 23.1 (121) | 25.2 (132) | 24.3 (93) | 27.7 (106) |
| **Clinician knowledge and communication** | | | | |
| Clinicians who understand autism | 10.7 (57) | 87.6 (467) | 16.0 (64) | 80.5 (322) |
| Opportunity after the appointment to ask questions about conclusions | 31.1 (166) | 60.2 (321) | 25.4 (100) | 64.4 (253) |
| Appointments at an easily identified and accessible location | 19.6 (104) | 70.4 (374) | 17.1 (70) | 71.7 (281) |
| Appointments with an easily identified and familiar clinician | 21.1 (112) | 68.9 (365) | 19.8 (78) | 71.0 (279) |
| Having a health summary document which can be shared with clinicians (eg, hospital passport) | 28.9 (152) | 51.0 (268) | 26.0 (100) | 48.8 (188) |
| A clinician who uses an approach which is informed by what you have said that you prefer (eg, formal or informal) | 31.6 (167) | 60.2 (318) | 30.6 (118) | 50.0 (193) |
| Identifying reasons that make it difficult to see a clinician or attend an appointment | 33.1 (174) | 49.5 (260) | 29.8 (114) | 52.9 (202) |
| **Stand-alone item** | | | | |
| Short waiting times to be seen when you attend appointments | 32.5 (168) | 55.1 (293) | 28.2 (111) | 57.8 (227) |

adjustment most frequently rated as 'never available' was 'being provided with appropriate distractions while waiting for appointments' and the item least frequently rated as 'never available' was 'appointments at an easily identified and familiar location'. A significant proportion of respondents to both surveys did not feel that they were able to comment on the availability of adjustments (range=mental health: 42.5%–62.4%; physical health: 19.4%–47.2%).

### Other adjustments endorsed by autistic people

Sixteen per cent (n=85) of participants in the mental health survey and 14% (n=56) in the physical health survey gave information in open text about an 'other' adjustment; some of these were not coded (eg, general comments). The majority (90% mental health, 91% physical health) of open text comments were classified within the three-factor solution (see table 4). An additional adjustment 'Being able to bring a supporter such as a friend, family member (or support animal) to appointments' was suggested by several participants in both health service contexts.

### Paired comparison of the importance and availability of adjustments between physical and mental health services

A total of 221 participants completed both surveys. After Bonferroni correction, only one adjustment was rated as being very important significantly more often in mental health (65%) compared with physical health settings (47%): 'a clinician who uses an approach which is informed by what you have said that you prefer' (p=0.001). No significant differences in the least available adjustments based on service type were observed. Full reporting

**Table 3** The lack of availability of key adjustments to mental and physical healthcare services to meet the needs of autistic adults

| Key adjustment | Mental health services | | Physical health services | |
|---|---|---|---|---|
| | Rarely available % (N) | Never available % (N) | Rarely available % (N) | Never available % (N) |
| **Sensory environment** | | | | |
| Change the sensory environment in the building that the appointment will take place in | 26.6 (63) | 45.6 (108) | 23.1 (55) | 60.5 (144) |
| Locations (eg, waiting rooms) with small numbers of people | 22.3 (63) | 29.3 (83) | 27.0 (72) | 49.1 (131) |
| Locations with low noise levels | 26.0 (72) | 26.0 (72) | 26.8 (73) | 42.6 (116) |
| Locations with low light levels | 24.3 (60) | 40.9 (101) | 30.9 (76) | 52.4 (129) |
| **Clinical and service context** | | | | |
| Changing the length of appointments to suit you | 23.1 (52) | 45.3 (102) | 29.3 (68) | 41.4 (96) |
| Offering appointments online or via apps | 16.4 (32) | 48.2 (94) | 17.6 (41) | 32.6 (76) |
| Changing how often you are asked to attend appointments | 25.4 (46) | 27.6 (50) | 23.1 (34) | 44.2 (65) |
| Give information to the clinician preappointment so that they can prepare | 22.8 (50) | 33.3 (73) | 25.1 (56) | 41.3 (92) |
| Provide support in relation to attending appointments (eg, managing fears or uncertainties which might make attending difficult) | 21.7 (51) | 40.4 (95) | 30.7 (67) | 45.9 (100) |
| Appropriate distractions provided while waiting to be seen at appointment (eg, tablet with headphones) | 15.7 (36) | 63.3 (145) | 18.3 (38) | 66.3 (138) |
| **Clinician knowledge and communication** | | | | |
| Clinicians who understand autism | 40.5 (105) | 31.3 (81) | 36.1 (84) | 36.9 (86) |
| Opportunity after the appointment to ask questions about conclusions | 23.9 (63) | 18.9 (50) | 20.2 (52) | 26.8 (69) |
| Appointments at an easily identified and accessible location | 15.3 (47) | 10.1 (31) | 12.0 (55) | 13.0 (38) |
| Appointments with an easily identified and familiar clinician | 18.2 (52) | 13.0 (37) | 25.3 (73) | 16.1 (47) |
| Having a health summary document which can be shared with clinicians (eg, hospital passport) | 17.7 (34) | 52.1 (100) | 17.9 (33) | 63.6 (117) |
| A clinician who uses an approach which is informed by what you have said that you prefer (eg, formal or informal) | 27.9 (70) | 23.5 (59) | 20.8 (45) | 42.1 (91) |
| Identifying reasons that make it difficult to see a clinician or attend an appointment | 25.9 (55) | 34.0 (72) | 26.3 (52) | 46.5 (92) |
| **Standalone item** | | | | |
| Short waiting times to be seen when you attend appointments | 25.1 (72) | 27.2 (78) | 33.9 (97) | 35.7 (102) |

of these analyses can be found in online supplemental tables 7 and 8.

## DISCUSSION

This large study successfully identified for the first time autistic adults' views about the importance and availability of key adjustments that UK and international guidelines and research consider could improve accessibility and acceptability of healthcare services. Autistic adults strongly endorsed the importance of these adjustments; the relative importance of each adjustment was broadly equivalent across physical and mental health service contexts. However, autistic people experienced limited availability to key adjustments across both healthcare contexts—showing clear unmet clinical need. Given that autistic people are at increased risk of combinations of mental and physical health conditions, healthcare providers need to take action now to implement provision of adjustments to improve service accessibility. Furthermore, this UK sample was large, recruited nationwide and was demographically diverse; as the majority of respondents reported they experienced both mental health conditions and physical health disorders, it is likely that these findings can be generalised to other autistic adults in the UK. This suggestion is further supported by the finding that there were no significant differences between the key demographic variables of responders and non-responders to the survey, with the exception of age in the physical health study (see online supplemental table 1). The availability and lack of availability of adjustments will, however, vary, not only within the UK, but also in other countries and especially those without publicly funded healthcare.

Three distinct but intercorrelated factors of adjustments were identified: (1) the sensory environment, (2) the clinical and service context and (3) clinician knowledge and understanding. These factors are consistent with recent syntheses of the evidence regarding the barriers to healthcare access for autistic people[12 13] and recent NHS reasonable adjustment guidance,[26] which include adjustments under the categories of environment, service delivery and communication. Our data-driven approach to categorisation is novel. The findings support and extend previous theoretical attempts at categorisation of adjustments to healthcare. The majority of 'other' adjustments proposed by participants could be classified within the three-factor model, providing further validation. Participants also identified across both service contexts the benefit of having the opportunity to attend appointments with support (a friend, family member). In common with all the adjustments included in our survey, this adjustment is not on the face of it difficult to implement, and it is likely most services would be supportive if asked. The NHS reasonable adjustments policy does already include having support at appointments among the potential adjustments offered[26] and recommends that such a request should be discussed and noted.

The adjustment rated as being most important in both settings was 'access to a clinician who understands autism'. Indeed, the four most important adjustments endorsed across both healthcare settings related to the 'clinician knowledge and communication' factor. This finding is in keeping with previous findings that healthcare professionals may lack skills and/or confidence in delivering interventions to autistic people.[17] A recent review found that some autistic people feel reluctant to report their autism diagnosis to healthcare professionals for fear that it may negatively affect their care.[35] This suggests that care must be taken to establish an environment whereby the patient feels confident enough to discuss their autism and what adjustments would help them fully access healthcare services openly. When the priority adjustments were compared across the paired sample, only one adjustment 'A clinician who uses an approach which is informed by what you have said that you prefer (eg, formal or informal)' was significantly different between healthcare contexts. Perhaps there is a perception that mental healthcare clinicians need to take into account individual preference; however, coming to a shared understanding and approach to healthcare is critical in any clinical setting. The discrepancy between the rated importance of the identified adjustments and their reported availability in both physical and mental health services makes stark reading. Our findings highlight the wide 'gap' in the perceived availability of adjustments reported as important. Closer examination of our findings on the availability of adjustments allows us to highlight potential shortfalls in UK health services. The high percentages of 'rarely or never available' responses seen across many adjustments suggests a systematic shortfall across the NHS rather than variability between services.

**Table 4** The number of open text comments that could be categorised to the identified factors following framework analysis

| | Mental health services | Physical health services |
| --- | --- | --- |
| Sensory environment | 8 | 5 |
| Clinical and service context | 33 | 24 |
| Clinician knowledge and communication | 16 | 10 |
| Other | 6 | 4 |

However, encouragingly two adjustments (appointments at a familiar location and with a familiar clinician) were rated as being available more than rarely by over half of each sample. Offering consistency in clinical care was appreciated in both contexts. In keeping with recently published findings identifying widespread barriers to healthcare access for autistic people,[12 13] respondents consistently rated their experiences of not being able to access adjustments in healthcare. If autistic adults are not experiencing these adjustments to promote accessibility, they are likely to continue to experience delays in mental and physical health diagnosis, poor treatment outcomes,[20] significant inequalities and premature mortality.[11]

### Strengths and limitations

Our study has many strengths. The research aimed to match published autism community research priorities and focus directly on the experiences of autistic people.[18] The study also includes experiences about adjustments in both mental and physical healthcare settings. Our large samples were both approximately equal in terms of male and female participants, which is a strength considering the low numbers of female participants often observed in autism research. Our data-driven approach enabled us to determine key overarching factors or categories of adjustments that could be used to inform both policy and practice in relation to service development.

One limitation of our study is that some participants reported that they did not have sufficient knowledge to comment on the availability of adjustments and were not able to provide a rating. However, the use of paired comparisons gives confidence that any identified differences are a true reflection of differences between mental and physical healthcare by virtue of eliminating the potential for sources of between-person variance. Further, that a notable proportion of the sample reported they were not informed of the availability of adjustments they might want, is an important finding in itself. Our data are based on the reported experiences of the respondents. We did not aim to collect information directly from the services, and this should be done in future studies. However, the samples are large and recruited from diverse sources across the UK including the NHS, private and charity providers, so it is unlikely systematic bias was introduced.

The participants in the mental health sample were all individuals who had self-reported either a diagnosed or suspected anxiety condition (relating to the aims of the original study). This may have the potential to limit the generalisability of the findings for other mental health settings—although many mental health services have a similar service design, and clinicians tend to deliver care for a variety of mental health conditions.

## Implications and future research

The findings of this study highlight the lack of key adjustments available to autistic adults currently accessing physical and mental healthcare. These adjustments do not require expensive technology or equipment but rather ongoing staff training, attention to aspects of the clinical settings and flexibility in ways of delivering clinical care. We consider that 'how' and 'what' adjustments can be made should be more explicit in pre-appointment information, and during appointments. Focusing on the sensory environment, clinical and service context, clinician knowledge and communication and involvement of supporters, may help service users, providers and commissioners of services to identify and address any barriers to implementation. Proactively considering these adjustments with autistic people is in our opinion a prudent and logical first step towards minimising the barriers to healthcare. This in turn is a crucial component of an individualised/personalised approach to assessment and management of identified healthcare needs in order to maximise treatment effectiveness and reduce health inequalities. Our current research is developing and evaluating personalised mental and physical healthcare provision for autistic people across the lifespan. Examples include the codesign (with autistic people, relatives and clinicians) and evaluation of a personalised, flexible psychological intervention for anxiety in autistic adults (PAT-A).[36] We have also codesigned an intervention to investigate how autistic people aged 50 and over can be encouraged to access the healthcare they need (Trial registration: ISRCTN 75726745). To address the primary care needs of autistic adults, we have codesigned and are evaluating the use of primary care healthchecks for autistic people (https://research.ncl.ac.uk/autismhealthchecks/). All these approaches systematically gather information about the adjustments required to maximise the chances of success (details of the tools used to do this available on request). Codesigned approaches to healthcare that meets the needs of autistic people and clinicians should increase the opportunities for autistic people to access the personalised and precision healthcare that is increasingly available to all people in mental and physical health services. In turn, that will hopefully reduce the increased and premature morbidity and average earlier mortality for autistic people.

**Acknowledgements** We are grateful to: all the participants who gave their time to complete the PAT-A and IHOAP surveys; the IHOAP investigators; the ASC-UK team including Dr Alex Petrou (who supported data collection), and the ASC-UK investigators, including Professor Helen McConachie. The Sponsor for the studies was Cumbria, Northumberland, Tyne and Wear NHS Foundation Trust.

**Contributors** JRP, BI and JR conceptualised the adjustments study. JRP and JR received funding for PAT-A. BI and JRP received funding for IHOAP. JRP and BI liaised with NHSE about adjustments. JR and MF supervised statistical analyses. SB and DM are researchers for PAT-A and IHOAP, respectively, and led data collection and undertook analysis. CW, MF and ALC provided comment on PAT-A and adjustment study design and analyses. All authors contributed to adjustments study design, and writing of the manuscript and reviewed the final version.

**Funding** Data from autistic adults were collected as part of two studies: 1. Personalised Anxiety Treatment for Autistic Adults study funded by the UK autism research charity Autistica. Funding was awarded to JRP and JR as part of the autism spectrum adulthood and ageing research programme at Newcastle University (Grant ID: 7425). 2. Improving the Health of Older Autistic People study, funded by the Inge Wakehurst Trust, awarded to BI and JRP (no grant number given).

**Competing interests** MF reports personal fees from Honoraria from training organisations in the area of psychological therapies, personal fees from Royalties from books in the area of psychological therapies, outside of the submitted work. All other authors declare no conflicts of interest.

**Patient consent for publication** Not required.

**Ethics approval** All procedures were approved by the NHS Health Research Authority and a Research Ethics Committee (both Wales REC 5, Approval reference numbers: PAT-A: 18/WA/0014; IHOAP: 18/WA/0191).

**Provenance and peer review** Not commissioned; externally peer reviewed.

**Data availability statement** Data are available on reasonable request. Some data that support the findings of this study are available on reasonable request from the corresponding author, (JRP). The data are not publicly available due to containing information that could compromise the privacy of research participants.

**ORCID iD**
Samuel Brice http://orcid.org/0000-0002-3501-2752

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
