## [Reviewer comments · BMJ Open]

ARTICLE DETAILS

TITLE (PROVISIONAL)	The importance and availability of adjustments to improve access for autistic adults who need mental and physical healthcare: findings from a UK survey
AUTHORS	Brice, Samuel; Rodgers, Jacqui; Ingham, Barry; Mason, David; Wilson, Colin; Freeston, Mark; Le Couteur, Ann; Parr, Jeremy

VERSION 1 – REVIEW

REVIEWER	Jennifer Ames Kaiser Permanente Northern California Division of Research, USA
REVIEW RETURNED	28-Aug-2020

GENERAL COMMENTS	Overview: This study reports results from two surveys administered to autistic adults regarding the importance and availability of different types of accommodations in primary care and mental health settings. The surveys were identical but were administered in two different study samples, focused on anxiety and physical health respectively, that were drawn from the Adult Autism Spectrum Cohort in the UK. There were some overlapping respondents across the two surveys. Survey respondents rated multiple adjustments across categories of sensory environment, clinical and service context, and clinician knowledge and communication as important and infrequently available. These findings are useful for healthcare providers and policymakers in prioritizing and targeting adjustments to make healthcare more accessible for autistic adults in the UK. This study is greatly enhanced by the thoughtful involvement of autistic people throughout the research process. Further, the manuscript is very well-written and well-organized. Below I offer a few minor edits and suggestions to clarify the implications of these findings. 1. In a couple places in the introduction, the authors summarize the higher likelihood of health conditions in autistic adults. It would be helpful to specify the reference group in these earlier studies (e.g., general population, non-autistic adults, etc).2. In the sentence “Evidence suggests that adjustments to services do facilitate healthcare access for autistic people (13).” Could the authors provide an example of 1-2 specific adjustments highlighted in this earlier work? Up until this point, the introduction has mostly focused on specific barriers so offering an example of a facilitator will help paint a fuller picture for the reader.3. The rationale for aim 2, which applied exploratory factor analysis to identify meaningful groupings of accommodations, could be made clearer. For example, in the physical health survey, the adjustment “therapists who understand autism” didn’t load on to the factor designated “Clinician Knowledge and Communication” which seemed counterintuitive. Do these findings have implications for next steps of prioritizing and implementing these adjustments? Perhaps it would clarify if the authors’ described their hypotheses for this subanalysis.
---

	4. The gender ratio of respondents (approx. 50:50 women:men) was interesting given that autism affects men more frequently than women. Did the authors explore any differences in responses by gender? Given that rates of co-occurring conditions and healthcare utilization patterns differ by gender, there might be important differences in how male and female respondents rate the adjustments. 5. The abstract and discussion report on the finding that “access to a clinician who understands autism” was rated as most important adjustment in both settings. Given the importance of this finding, it should be described in the text of the results section as well. 6. The limitations section mentions that a sizable proportion of respondents couldn’t comment on the availability of some adjustments. The magnitude of this proportion should be reported in the results section. 7. Given the characteristics of those who responded to the survey, could the authors discuss the generalizability of these findings to adults across the autism spectrum in the UK? For example, in terms of the sociodemographic diversity, the age at ASD diagnosis, support systems, etc. 8. What is the difference between the results reported in Supplementary Tables 5 and 6? The numbers are slightly different but the table titles are the same. Minor revisions: 9. In the results section of the abstract, it is unclear if the numbers in parentheses are the two p-values from the separate surveys? I do not think it is essential to report the p-value here so you could consider removing it. 10. “Several UK autism community top research priorities” was a bit wordy. Could this be rephrased? 11. The sentence on page 8, “Prior to commencing both studies... to discuss and agree the methods and materials.” Is missing a “to” after agree. 12. On page 11, for the sentence “In the physical health, but not the mental health study, responders were significantly older than nonresponders (t(668)= 3.369, p=0.001)”, I suggest replacing the t-statistic with the average ages in the two samples. This will be more readily interpretable for most readers. 13. In the anxiety survey sample, the authors report on the proportion of respondents who had completed, stopped, or never started psychological services. Did they have information on whether any of the respondents were currently in treatment? 14. One of the strengths raised in the discussion is “the use of paired comparisons provides more statistical power than between group analyses.” This point is not demonstrated in the results but perhaps there is a relevant reference that can be included here or in the methods section? 15. Supplementary table 1 is missing some proportions (for example gender in non-responders).
--	--

REVIEWER	Brittany Hand The Ohio State University, USA
REVIEW RETURNED	31-Aug-2020

GENERAL COMMENTS

Overall

- This is an incredibly important topic. Thank you for conducting this valuable work. The stated objectives of the study were: (1) To investigate autistic people's views on the importance and availability of adjustments that are designed to promote the accessibility and acceptability of mental and physical healthcare provision, (2) To explore whether specific categories of adjustments can be identified using exploratory factor analysis and to identify any differences in their importance and availability between mental and physical healthcare.
- There is no dispute here about the importance of this work and the valuable contribution this project will make to the field. However, the manuscript would benefit from revision to better align the objectives, statistical analyses designed to achieve those objectives, and presentation of results. These three sections of the manuscript should be clearly linked to one another. The results section of the paper includes multiple hypothesis test results that do not clearly link to the stated objectives of the paper and were not described in the analysis section. The ordering of information that is presented in the analysis section does not parallel how the results are presented, which also detracts from the readability.

Abstract

- Lines 43-50 – Please rephrase to indicate the directionality of the difference. For example, “Only one significant difference between the importance of adjustments available through mental and physical health services was identified; participants reported mental health clinicians were significantly more willing to adapt their approach to the person's preferences ($p = .001 < .0028$).”

Introduction

- Page 5 lines 33-34 – “with as many as 12 years less life” is confusingly worded here. Does that mean that autistic people die, on average, 12 years earlier than non-autistic individuals?
- Page 6 lines 17-19 – please clarify “which may impact on engagement and retention.” Engagement with what/whom? Retention of patients, retention of providers, or both?
- The objectives/aims of this study are worded as exploratory and the second (re: EFA) is truly exploratory. However, there were multiple hypothesis tests conducted in this study. Did the authors have an a-priori hypothesis? If not, please justify the use of hypothesis testing for an exploratory objective (“to investigate autistic people's views...”).

Methods

- Page 11 line 35-43 – Is data about the individuals who were removed from the analysis prior to EFA included in the table of participant characteristics? If their data were not included in the analysis, then I would think they should not be included in the description of the sample.
- What are “service-level differences”? What process was used to identify adjustments that were “deemed most important to autistic people and least available”? After reading through the results section and circling back, I can tell that “service level differences” refers to comparisons between physical and mental health care but this was not apparent upon first reading through. Additionally, adjustments were “deemed most important and least available” based on the ratings of participants. This also was not readily apparent upon first read through.
- Please add to the supplemental methods what cut-off values you used for each fit index to determine goodness-of-fit.
- Please structure the analysis section so that you describe the analyses you will conduct in the same order that they will be presented in the results. Currently, the EFA is described first in the analysis section but reported near the end of the results.

	Results  • How can you include data about individuals who did not consent to be in your study? When individuals consented to be in the larger longitudinal study, did they also consent to have their data used for these types of comparisons? Please clarify. • Why were characteristics of the physical and mental health samples compared statistically? This doesn't seem to relate to your objectives and is not described in the analysis section. Please make sure that all statistical tests employed in this paper are described in the analysis section. • While the description of service use is interesting, it is not clear how this relates to the objectives of this study. This information could be summarized in a table rather than taking up valuable space in the manuscript. I would rather see the EFA methods brought back into the manuscript (rather than being supplemental) and this service use section cut since it is not directly linked to the objectives. • Page 15 – Please consider removing references to item numbers from the text as the item numbers were arbitrarily assigned and mean little to the reader. Instead, the authors may consider using item stems (e.g., “item 9” could be “change sensory environment”), which would help the reader understand which adjustments were rated as most important and least available without having to reference the tables. • Table 1 – Please add a footnote defining all abbreviations used within the table so that the table can stand alone. • Table 3 – Per the methods section, the categories here were: 1) never available, 2) rarely available, 3) available about half the time, 4) available most of the time, 5) available all of the time. It is unclear then why Table 3 presents “rarely or never available” and “never available” as the categories. Does this mean that all of the participants who responded “never available” are also represented again in the “rarely or never available” column? • Table 4 – It is unclear what the numbers in these tables are supposed to represent. Is that the number of open-ended comments that relate to those constructs? Please provide enough information so that the table can stand alone from the text. • Supplementary Table 1 – some percentages are missing from the “IHOAP non-responders” column Discussion  • Strengths and limitations appear to be appropriately discussed. • The discussion section may benefit from adding additional citations and discussion of the findings of this study in relation to other contemporary works. Currently, the citations included in the discussion section consist solely of works previously cited in the introduction.
--	---

VERSION 1 – AUTHOR RESPONSE

REVIEWER 1 COMMENTS

Overview: This study reports results from two surveys administered to autistic adults regarding the importance and availability of different types of accommodations in primary care and mental health settings. The surveys were identical but were administered in two different study samples, focused on anxiety and physical health respectively, that were drawn from the Adult Autism Spectrum Cohort in the UK. There were some overlapping respondents across the two surveys. Survey respondents rated multiple adjustments across categories of sensory environment, clinical and service context, and clinician knowledge and communication as important and infrequently available. These findings

are useful for healthcare providers and policymakers in prioritizing and targeting adjustments to make healthcare more accessible for autistic adults in the UK. This study is greatly enhanced by the thoughtful involvement of autistic people throughout the research process. Further, the manuscript is very well-written and well-organized. Below I offer a few minor edits and suggestions to clarify the implications of these findings.

Thank you for your kind words and helpful suggestions. We have responded below to each of the points raised.

1. In a couple places in the introduction, the authors summarize the higher likelihood of health conditions in autistic adults. It would be helpful to specify the reference group in these earlier studies (e.g., general population, non-autistic adults, etc).

Thank you for this comment. We have specified the reference group (general population

/ neurotypical adults) in studies that describe higher incidence of health conditions in autistic adults (page 4, lines 68-71).

2. In the sentence “Evidence suggests that adjustments to services do facilitate healthcare access for autistic people (13).” Could the authors provide an example of 1-2 specific adjustments highlighted in this earlier work? Up until this point, the introduction has mostly focused on specific barriers so offering an example of a facilitator will help paint a fuller picture for the reader.

We agree that this information would be helpful, and it has now been added to the introduction (page 5, lines 107-109; see below).

‘Evidence suggests that adjustments to services such as reducing exposure to potentially aversive sensory stimuli (e.g. noisy waiting rooms) and the clinician using the autistic adult’s preferred communication methods, do facilitate effective healthcare access for autistic people (13).’

3. The rationale for aim 2, which applied exploratory factor analysis to identify meaningful groupings of accommodations, could be made clearer. For example, in the physical health survey, the adjustment “therapists who understand autism” didn’t load on to the factor designated “Clinician Knowledge and Communication” which seemed counterintuitive. Do these findings have implications for next steps of prioritizing and implementing these adjustments? Perhaps it would clarify if the authors’ described their hypotheses for this subanalysis.

The rationale for the factor analysis was to enable us to empirically identify categories of adjustments derived from the survey data as a means to organise our results. Consistent with this approach we undertook data driven, exploratory factor analyses rather than testing apriori hypotheses. This approach provides a parsimonious way to group and then communicate key adaptations to service providers and clinicians. We have added some further information to the manuscript (page 5, lines 85-87; see below) and hope makes this rationale is clearer to the reader.

‘Although such attempts to categorise adjustments in this way are useful to provide a simplified message to healthcare providers, this categorisation system has yet to receive empirical support’

4. The gender ratio of respondents (approx. 50:50 women:men) was interesting given that autism affects men more frequently than women. Did the authors explore any differences in responses by gender? Given that rates of co-occurring conditions and healthcare utilization patterns

differ by gender, there might be important differences in how male and female respondents rate the adjustments.

Thank you for this suggestion. Exploration of gender identity-based differences in required adjustments was not a primary goal of the current study. We agree with the reviewer that this would be an interesting and important topic to explore in future research. It would be important for a future study to ensure that autistic people of all genders (not only males and females) were consulted regarding the content of the survey. As this was not a process undertaken in relation to the present study, we have not reported findings for males and females within the current study.

The abstract and discussion report on the finding that “access to a clinician who understands autism” was rated as most important adjustment in both settings. Given the importance of this finding, it should be described in the text of the results section as well.

Thank you for this suggestion. We have added a description to the results section (page 18, lines 352-354; see below).

‘Autistic adults rated ‘having access to a clinician who understands autism’ as the most important adjustment in both mental health (98.3%, n= 524) and physical health (96.5%, n= 386) settings.’

6. The limitations section mentions that a sizable proportion of respondents couldn’t comment on the availability of some adjustments. The magnitude of this proportion should be reported in the results section.

Thank you. We have added the range of proportions of respondents who were not able to comment on the availability of adjustments in the results section (page 20, line 386; see below).

‘A significant proportion of respondents to both surveys did not feel that they were able to comment on the availability of adjustments (range = mental health: 42.5% – 62.4%; physical health: 19.4% - 47.2%).’

7. Given the characteristics of those who responded to the survey, could the authors discuss the generalizability of these findings to adults across the autism spectrum in the UK? For example, in terms of the sociodemographic diversity, the age at ASD diagnosis, support systems, etc.

This suggestion was also made by the editorial team (point 10). Please see our response on page 2 of this letter.

8. What is the difference between the results reported in Supplementary Tables 5 and 6? The numbers are slightly different but the table titles are the same.

We appreciate that these tables do look similar and present similar data. Table 5 is the response frequencies for the importance of adjustments whereas Table 6 is the response frequencies for the availability of adjustments. The titles and column headings reflect these differences.

Minor revisions:

9. In the results section of the abstract, it is unclear if the numbers in parentheses are the two p-values from the separate surveys? I do not think it is essential to report the p-value here so you could consider removing it.

Thank you for raising this point. We had previously reported the P value for this comparison ($p = .001$) and the significance value following Bonferroni correction ($<$

.0028). We have now changed this to report only the p value, with further information regarding Bonferroni correction accessible in the results section of the main paper and in Supplementary Tables 7 & 8. We appreciate the point that you make about reporting

these figures in the abstract and may not ordinarily have chosen to do so, however this is a requirement of the journal.

10. "Several UK autism community top research priorities" was a bit wordy. Could this be rephrased?

We have rephrased this to read 'UK autism research priorities...' (page 5, line 98)

11. The sentence on page 8, "Prior to commencing both studies... to discuss and agree the methods and materials." Is missing a "to" after agree.

Thank you for pointing this out, we have added the missing word (page 8, line 181).

12. On page 11, for the sentence "In the physical health, but not the mental health study, responders were significantly older than nonresponders ($t(668) = 3.369, p = 0.001$)", I suggest replacing the t-statistic with the average ages in the two samples. This will be more readily interpretable for most readers.

Thank you for this suggestion. We would prefer to leave the t statistic in the manuscript but we have also added the mean ages in parentheses (page 12, line 269-270; see below).

'In the physical health, but not the mental health study, responders were significantly older (mean age = 44.4) than non-responders (41.0) ($t(668) = 3.369, p = 0.001$).'

13. In the anxiety survey sample, the authors report on the proportion of respondents who had completed, stopped, or never started psychological services. Did they have information on whether any of the respondents were currently in treatment?

Unfortunately, these data were not available to us in this study.

14. One of the strengths raised in the discussion is "the use of paired comparisons provides more statistical power than between group analyses." This point is not demonstrated in the results but perhaps there is a relevant reference that can be included here or in the methods section?

Thank you this suggestion. We have added further information to clarify, namely that paired samples offer greater statistical power by virtue of the fact that sources of between-person variance (i.e. confounding variables) at participant level is eliminated (page 24, line 490). We have also referenced the following work to illustrate this point.

Wacholder, S., & Weinberg, C. R. (1982). Paired versus two-sample design for a clinical trial of treatments with dichotomous outcome: power considerations. Biometrics, 801-812.

15. Supplementary table 1 is missing some proportions (for example gender in non-responders).

Thank you for highlighting this, we have corrected this error in supplementary Table 1

REVIEWER 2 COMMENTS

Please leave your comments for the authors below

Overall

This is an incredibly important topic. Thank you for conducting this valuable work. The stated objectives of the study were: (1) To investigate autistic people's views on the importance and availability of adjustments that are designed to promote the accessibility and acceptability of mental and physical healthcare provision, (2) To explore whether specific categories of adjustments can be identified using exploratory factor analysis and to identify any differences in their importance and availability between mental and physical healthcare.

There is no dispute here about the importance of this work and the valuable contribution this project will make to the field. However, the manuscript would benefit from revision to better align the objectives, statistical analyses designed to achieve those objectives, and presentation of results. These three sections of the manuscript should be clearly linked to one another. The results section of the paper includes multiple hypothesis test results that do not clearly link to the stated objectives of the paper and were not described in the analysis section. The ordering of information that is presented in the analysis section does not parallel how the results are presented, which also detracts from the readability.

Thank you for your kind words and helpful suggestions. We have responded to each point in turn below.

Abstract

1. Lines 43-50 – Please rephrase to indicate the directionality of the difference. For example, “Only one significant difference between the importance of adjustments available through mental and physical health services was identified; participants reported mental health clinicians were significantly more willing to adapt their approach to the person's preferences ($p = .001 < .0028$).”

We agree with your comments and have rephrased this sentence in the abstract in line with your suggestion (page 2, lines 38-40; see below).

'Participants reported that having access to a clinician who is willing to adapt their approach to suit the person's preferences was significantly more important for participants attending mental health settings (p = .001).'

Introduction

2. Page 5 lines 33-34 – “with as many as 12 years less life” is confusingly worded here. Does that mean that autistic people die, on average, 12 years earlier than non-autistic individuals?

Thank you for this suggestion, your understanding of this point is correct. We agree that the wording could be clearer and have amended the manuscript accordingly (page 4, lines 75-77; see below).

'Epidemiological studies suggest autistic people are also significantly more likely to face premature mortality: the average life expectancy for an autistic adult without intellectual disability is 12 years lower than for neurotypical people (11).'

3. Page 6 lines 17-19 – please clarify “which may impact on engagement and retention.”
Engagement with what/whom? Retention of patients, retention of providers, or both?

We refer to the idea that autistic people may be less likely to engage with healthcare services that are not sufficiently adapted to meet their needs. Our choice of wording could have illustrated the point more clearly and we have amended the manuscript as such (page 5 line 96; see below)

'There is also evidence of a lack of training, skills and/or confidence among healthcare professionals in working with and delivering interventions to autistic people (17), which may impact on their continued engagement with healthcare services (12).'

4. The objectives/aims of this study are worded as exploratory and the second (re: EFA) is truly exploratory. However, there were multiple hypothesis tests conducted in this study. Did the authors have an a-priori hypothesis? If not, please justify the use of hypothesis testing for an exploratory objective (“to investigate autistic people's views...”).

We agree that the aims of the study are exploratory and worded as such. We did not have any a priori directional hypotheses and for this reason wanted to investigate any potential differences between physical and mental health services through the use of two tailed hypothesis testing. Our rationale for this was to explore whether more bespoke advice surrounding adjustment implementation may need to be delivered depending on whether the

health service provided physical or mental healthcare. Other tests of significance were related to comparisons on key demographic variables between the two samples to aid with interpretation of the findings.

Methods

5. Page 11 line 35-43 – Is data about the individuals who were removed from the analysis prior to EFA included in the table of participant characteristics? If their data were not included in the analysis, then I would think they should not be included in the description of the sample.

Thank you for raising this point. Participants with one or more missing responses to the importance/availability of adjustments were not included in the factor analyses (i.e. deleted listwise). This reflects best practices in EFA. These participants did however respond to some importance/availability items (indeed many responded to most items) and were therefore included in subsequent analyses in this study. For this reason, these participants are included in the table of participant characteristics. For clarity, some additional participants (31 in the anxiety survey, 54 in the physical health survey; reported on page 7, lines 148-150, 157-158) consented but did not respond to any of the adjustment items and these participants are not included in the demographic table.

6. What are “service-level differences”? What process was used to identify adjustments that were “deemed most important to autistic people and least available”? After reading through the results section and circling back, I can tell that “service level differences” refers to comparisons between physical and mental health care but this was not apparent upon first reading through. Additionally, adjustments were “deemed most important and least available” based on the ratings of participants. This also was not readily apparent upon first read through.

Thank you for raising this point. We agree that ‘service level differences’ was not adequately described and we have added an additional description in parenthesis to rectify this (pages 11, line 249). We think that the existing description in the analysis section on page 11 (lines 252-255), clearly describes the procedure used to dichotomize the variables to determine ‘most important’ and ‘least available’.

7. Please add to the supplemental methods what cut-off values you used for each fit index to determine goodness-of-fit.

We agree that this is useful information and it has now been included in the supplementary materials underneath Supplementary Table 4 (see below).

The following cut-off values were used to determine acceptable goodness of fit:

RMSEA < 0.08

NNFI ≥ 0.95

CFI \geq 0.90

GFI \geq 0.95

RMSR $<$ 0.08.

8. Please structure the analysis section so that you describe the analyses you will conduct in the same order that they will be presented in the results. Currently, the EFA is described first in the analysis section but reported near the end of the results.

We agree with your comments. We have changed the structure of the analysis section so that it now matches the presentation of the results and describes all analyses undertaken. This has improved the clarity and flow of the manuscript (pages 10 & 11)

Results

9. How can you include data about individuals who did not consent to be in your study? When individuals consented to be in the larger longitudinal study, did they also consent to have their data used for these types of comparisons? Please clarify.

When people join the Autism Spectrum Cohort UK, they give permission for us to use the information they have provided in our research. We have previously published papers from the cohort showing the characteristics of responders and non-responders as this is considered best practice in large studies where anonymous data are available (see McConachie et al., 2018 as an example)

McConachie, H., Mason, D., Parr, J. R., Garland, D., Wilson, C., & Rodgers, J. (2018). Enhancing the validity of a quality of life measure for autistic people. *Journal of Autism and Developmental Disorders*, 48(5), 1596-1611.

10. Why were characteristics of the physical and mental health samples compared statistically? This doesn't seem to relate to your objectives and is not described in the analysis section. Please make sure that all statistical tests employed in this paper are described in the analysis section.

The demographic characteristics of the physical and mental health sample were compared statistically in order to demonstrate their comparability. This is important given the significant heterogeneity of the autism spectrum and because the data originate from two distinct studies. The finding that there are no significant differences allows more confidence when reporting the results on the importance and availability of adjustments. If we had found widely differing results between the samples, it would be difficult to tell whether these were due to the types of service or to the samples themselves. We have added further detail to the analysis section (page 10, lines 219-224; see below) to ensure that all statistical tests are adequately described.

'Key demographic characteristics (age, gender and level of autism traits) were compared using independent t-tests or ANOVA for participants who self-reported an autism diagnosis, and those who suspected that they were autistic or were awaiting an autism diagnosis; responders vs non-responders (in both studies) and between participants from the mental health and physical health samples. These comparisons were made to assess the representativeness and comparability of the samples and to aid with interpretation of the findings.'

11. While the description of service use is interesting, it is not clear how this relates to the objectives of this study. This information could be summarized in a table rather than taking up valuable space in the manuscript. I would rather see the EFA methods brought back into the manuscript (rather than being supplemental) and this service use section cut since it is not directly linked to the objectives.

We agree that the rationale for looking at data regarding service use could be made clearer in the methods section and have attempted to do so (page 10, lines 226-229; see below). The results presented relate directly to the objectives of the study by providing evidence that a) adjustments are not being routinely offered and b) that this is a potential cause of autistic adults disengaging with services. The EFA analyses were a preliminary step to facilitate organisation of the survey items into empirically derived factors which were used to present the results. As this procedure was not related to the objectives of the study, we consider that the EFA methods and findings are appropriately placed in the supplemental files.

'Descriptive analyses of survey responses regarding prior service use in both samples were undertaken to provide further context. This information included the proportion of participants who had previously accessed mental and physical healthcare, their engagement with treatment and preliminary indications regarding their experience of adjustments to healthcare.'

12. Page 15 – Please consider removing references to item numbers from the text as the item numbers were arbitrarily assigned and mean little to the reader. Instead, the authors may consider using item stems (e.g., “item 9” could be “change sensory environment”), which would help the reader understand which adjustments were rated as most important and least available without having to reference the tables.

We have removed the item numbers from the revised manuscript and added more descriptive information about the nature of the adjustment.

13. Table 1 – Please add a footnote defining all abbreviations used within the table so that the table can stand alone.

Thank you for this suggestion, we agree that this improves the interpretability of the table and we have added a footnote defining all abbreviations (page 15).

14. Table 3 – Per the methods section, the categories here were: 1) never available, 2) rarely available, 3) available about half the time, 4) available most of the time, 5) available all of the time. It is unclear then why Table 3 presents “rarely or never available” and “never available” as the categories. Does this mean that all of the participants who responded “never available” are also represented again in the “rarely or never available” column?

When there are five response categories, it can be difficult to get a clear sense of the differential patterns across multiple items and in this case also across two different settings (and we have made the full response frequencies available in supplementary Tables 5 & 6). We examined the raw data in depth before making the following choices about how to present the data. We have chosen to present the data first as ‘rarely or never available’ (i.e. those who selected option 1 OR option 2) as representing “to all practical purposes, not available” versus the other percentage (implied, but not reported) of “available at least half of the time or more”. This is then easily comparable vertically and horizontally. The second column, ‘never’ then represents adjustments that, in the participants view, are “in absolute terms, not available”. Together, we believe these two figures provide the reader with a clearer sense of unavailability of these adaptations: the higher the “never” figure, and the more approaches the “to all practical purposes, unavailable”, the greater the evidence of a system-wide lack of adjustment, rather than variability between different services (see additions to the discussion, page 23, lines 469-472). The alternative, we believe, of providing two or more categories separately would require greater synthesis by the reader.

15. Table 4 – It is unclear what the numbers in these tables are supposed to represent. Is that the number of open-ended comments that relate to those constructs? Please provide enough information so that the table can stand alone from the text.

These figures do represent the number of open-ended statements that relate to these constructs/factors. We have added additional information to the title of Table 4 to improve interpretability (page 20).

16. Supplementary Table 1 – some percentages are missing from the “IHOAP non-responders” column

Thank you, we have now added percentages that were previously missing from Supplementary Table 1.

Discussion

Strengths and limitations appear to be appropriately discussed.

17. The discussion section may benefit from adding additional citations and discussion of the findings of this study in relation to other contemporary works. Currently, the citations included in the discussion section consist solely of works previously cited in the introduction.

Thank you for this suggestion. This is a relatively new research field and there is a relative scarcity of contemporaneous research to discuss in relation to our findings. Much of the research in this area to date has focussed on identifying barriers to

healthcare, best summarised in two very recent reviews that we have cited. We have added another sentence that attempts to better link our findings to other work (page 22, lines 435-437) and another reference (below) that offers another important consideration (page 22-23, lines 450-455).

Walsh, C., Lydon, S., O’Dowd, E., & O’Connor, P. (2020). Barriers to Healthcare for Persons with Autism: A Systematic Review of the Literature and Development of A Taxonomy. *Developmental Neurorehabilitation*, 1-18.

VERSION 2 – REVIEW

REVIEWER	Jennifer Ames Kaiser Permanente Northern California, USA
REVIEW RETURNED	19-Nov-2020

GENERAL COMMENTS	I thank the authors for their very thorough and thoughtful response to the first review. The revision reads well and the study’s motivation and implications have been adequately clarified. This study is an important contribution to the literature, highlighting unmet needs among autistic adults in both mental and physical healthcare and identifying healthcare adjustments of high priority. I just have very minor suggested edits.  1. Abstract: The three adjustment areas are referred to as “categories” in the results but “factors” in the conclusions. Same terminology should be used in both places for clarity. 2. Methods: The authors mention using t-tests and ANOVA to compare across groups. ANOVA is typically used to compare means across several groups. The authors may have meant to say that chi-square tests were used to compare the distribution of categorical variables across groups. 3. Please doublecheck the mental health sample sizes reported in the sentence: “In mental health services, those rating each adjustment as important ranged from 46% (n=240) (access to online appointments) to 98% (n= 525) (clinician who understands autism).” They differ slightly from Table 2. 4. As the authors point out, the paired analysis among participants who took both surveys does remove confounding by some measured and unmeasured factors. However, I am having difficulty following the discussion’s point about power (which is related more to statistical efficiency) since the authors did not demonstrate that the two sample design (which is more than double the size of the paired design) was underpowered to detect a significant differences in adjustment importance by service-level. The loss in sample size in the paired analysis is important because the cited Wacholder reference suggests that statistical precision is greater in a paired design compared to a two sample design when the number of observations is the same. While the paired design may actually reduce power in the present study, the tradeoff is the ability to internally control for potential confounding factors and thus obtain a more accurate estimate of the differences between the service-levels. The authors may want to consider reframing the strengths of the paired design not in terms of power (random error) but in terms
--

	of bias reduction.
REVIEWER	Brittany Hand The Ohio State University, USA
REVIEW RETURNED	12-Nov-2020

GENERAL COMMENTS	Overall  • This is an incredibly important topic. Thank you for conducting this valuable work and for the opportunity to re-review the manuscript. The stated objectives of the study were: (1) To investigate autistic people's views on the importance and availability of adjustments that are designed to promote the accessibility and acceptability of mental and physical healthcare provision, (2) To explore whether specific categories of adjustments can be identified using exploratory factor analysis and to identify any differences in their importance and availability between mental and physical healthcare. • The authors have made substantial edits to this paper to better align the objectives, methods, and presentation of results. I still feel, however, that the paper would benefit from some re-organization and clarification around certain points (i.e., whether the authors are referring to the parent studies or the sample for the present study at various points; consistency in terms used to refer to samples and surveys). I also recommend the authors reduce the amount of space in the main manuscript that is dedicated to describing the study sample (currently 4 pages) to make room for moving findings related to the study objectives to the main manuscript instead of reporting in the supplement. Abstract  • Prior comments were adequately addressed. Introduction  • Prior comments were adequately addressed. Methods  • Line 209 – it is unclear why the SRS-2 is described under its own sub-heading within the methods section and is not included under the “measures” sub-section. • Page 10 line 226 and 232 (per the author’s line numbering) - it is unclear what “both samples” and “both datasets” refer to. Does this mean responders and non-responders? Does this mean participants with suspected or confirmed autism diagnosis? Does this mean participants who were recruited from the PAT-A or IHOAP study? If the latter, what was done about the participants in this study who participated in both the PAT-A and IHOAP studies – were their data included only once? It becomes clear around lines 237-238 that “both” refers to a “mental health dataset” and “physical health dataset.” I am guessing that these are determined based on whether the participant was recruited from the PAT-A or IHOAP study, but this is not explicitly stated. • Line 243 – is this supposed to say “qualitative data”? The rest of this paragraph is about the qualitative data coding. • Line 245-246 – how can there be discrepancies if there was only one researcher who coded the data? • Please clarify whether the EFA was conducted on both samples separately or if they were combined. It appears from the results that they were analyzed separately, but that should be explained in the
---

	methods. Results  • Line 260-264 – was this analysis done on the full PAT-A and IHOAP samples or only those individuals who participated in this study? I think this is unclear because of the order in which information is presented in the results section. The following paragraph talks about responders vs non-responders. If the paragraph about responders vs. non-responder was first and the paragraph about the SRS scores came after, the SRS scores paragraph could begin with “Among respondents in both the mental and physical health samples...” if indeed this SRS score comparison was conducted only on responders for this study. I think this would also be clearer if the sample size and response rate information was reported in the results, with the information about responders and non-responders, rather than the methods (lines 146-150 and lines 157-162). • Lines 288 – I would recommend choosing whether you want to refer to this as a “mental health survey” or “anxiety survey” and being consistent throughout. If you want to refer to it as an “anxiety survey” then the sample should also be described as such instead of the “mental health sample.” • Lines 288-307 – it is unclear if these two samples were sent out different surveys for the present study, or if the information about medication and service use was pulled from the parent studies from which the participants in this study were recruited. If they were sent out distinct surveys for this study, then that should be clarified in the methods section. If this data was obtained from the parent studies from which these individuals were recruited, that should be clarified in lines 288-307. • Previous comment that was not adequately addressed in this revision – “While the description of service use is interesting, it is not clear how this relates to the objectives of this study. This information could be summarized in a table rather than taking up valuable space in the manuscript. I would rather see the EFA methods brought back into the manuscript (rather than being supplemental) and this service use section cut since it is not directly linked to the objectives.” I understand that you want to include this to provide context about the individuals who participated in the study, but this could be summarized in a table -- it could probably be easily incorporated into Table 1. Currently, there 4 full pages of the manuscript (pages 12-15) dedicated to describing the sample rather than using that valuable space to address the stated objectives of the study. It certainly is important to describe the sample, but (in my opinion) too much space has been dedicated to this. As a result, findings that are more directly related to your objectives have been placed in the supplemental materials instead of the main manuscript. • Previous comment that was not adequately addressed – “Table 3 – Per the methods section, the categories here were: 1) never available, 2) rarely available, 3) available about half the time, 4) available most of the time, 5) available all of the time. It is unclear then why Table 3 presents “rarely or never available” and “never available” as the categories. Does this mean that all of the participants who responded “never available” are also represented again in the “rarely or never available” column?” -- It would make more sense to me to have the columns be “rarely available” and “never available”, that way, if readers want to sum them up, they can
--	--

	and there would not be responses that are included in both columns. Discussion  • Prior comments were adequately addressed.
--	--

VERSION 2 – AUTHOR RESPONSE

Reviewer: 2

Comments to the Author

Overall

- This is an incredibly important topic. Thank you for conducting this valuable work and for the opportunity to re-review the manuscript. The stated objectives of the study were: (1) To investigate autistic people’s views on the importance and availability of adjustments that are designed to promote the accessibility and acceptability of mental and physical healthcare provision, (2) To explore whether specific categories of adjustments can be identified using exploratory factor analysis and to identify any differences in their importance and availability between mental and physical healthcare.

Thank you for your helpful comments and suggestions.

Abstract

- Prior comments were adequately addressed.

Introduction

- Prior comments were adequately addressed.

Methods

- Line 209 – it is unclear why the SRS-2 is described under its own sub-heading within the methods section and is not included under the “measures” sub-section.

We agree and we have removed the subheading for the SRS-2 so that it now sits under the broader ‘measures’ subheading.

- Page 10 line 226 and 232 (per the author’s line numbering) - it is unclear what “both samples” and “both datasets” refer to. Does this mean responders and non-responders? Does this mean participants with suspected or confirmed autism diagnosis? Does this mean participants who were recruited from the PAT-A or IHOAP study? If the latter, what was done about the participants in this study who participated in both the PAT-A and IHOAP studies – were their data included only once? It becomes clear around lines 237-238 that “both” refers to a “mental health dataset” and “physical health dataset.” I am guessing that these are determined based on whether the participant was recruited from the PAT-A or IHOAP study, but this is not explicitly stated.

Thank you for highlighting this, we have made the following changes that hopefully make this clearer:

We have changed ‘in both studies’ to ‘in both the mental health and physical health studies’ (page 10, lines 223-4)

We have also added (moved from the results section) the following sentence to the first paragraph of the ‘analysis’ section of the method:

“Participants who completed both studies were not included in any statistical comparisons made between the mental health and physical health samples to avoid double counting and so breaking the assumption of independence.” (page 10, lines 226-8)

The reference to ‘both samples’ is no longer in the manuscript due to changes to our reporting of ‘service use’ as per your suggestions in a subsequent point.

- Line 243 – is this supposed to say “qualitative data”? The rest of this paragraph is about the qualitative data coding.

The word quantitative is intentionally used in this instance to describe our quantitative analyses (i.e. descriptive). We then discuss the analysis of our open text data. We have reservations about using the word qualitative to describe this data as we do not consider our analysis to be truly qualitative. We appreciate that this transition may be difficult to follow but hope that inclusion of the wording below makes this clearer to the reader

“Quantitative data were analysed descriptively. Analysis of open text comments and descriptions of experiences about adjustments used framework analysis...” (page 11, lines 264-5).

- 2. Line 245-246 – how can there be discrepancies if there was only one researcher who coded the data?

We have amended the manuscript to read:

“Two researchers (JR and SB) coded open text responses against the adjustment categories indicated in EFA and any discrepancies were discussed and agreed upon.” (page 11, lines 265-7)

/ Please clarify whether the EFA was conducted on both samples separately or if they were combined. It appears from the results that they were analyzed separately, but that should be explained in the methods.

There is now further detail regarding EFA under the analysis subheading of the methods section (previously reported in supplementary materials) and this includes the following sentence that adds clarity:

'...was used to conduct EFA on data on the importance of each adjustment in mental health and physical health services separately.' (page 10, lines 231-3)

Results

3. Line 260-264 – was this analysis done on the full PAT-A and IHOAP samples or only those individuals who participated in this study? I think this is unclear because of the order in which information is presented in the results section. The following paragraph talks about responders vs non-responders. If the paragraph about responders vs. non-responder was first and the paragraph about the SRS scores came after, the SRS scores paragraph could begin with “Among respondents in both the mental and physical health samples...” if indeed this SRS score comparison was conducted only on responders for this study. I think this would also be clearer if the sample size and response rate information was reported in the results, with the information about responders and non-responders, rather than the methods (lines 146-150 and lines 157-162).

Thank you for raising this suggestion. In line with a subsequent point that you make about giving less detail on sample characteristics, the reporting of these comparisons in the manuscript has changed significantly and is primarily accessible in the supplementary materials. We have now simply reported the following in the main manuscript and think this is clearer.

“Details of statistical comparisons between the study participants and those who did not respond, and within and between subsamples (i.e. mental health and physical health samples) can be seen after Supplementary Table 1. Of note, there was no significant difference in autism characteristics (SRS-2 total score) between those who reported an autism spectrum diagnosis and those who suspected they were autistic or were awaiting diagnosis in all comparison groups. The only significant differences identified were that the physical health sample (mean age= 44.4, SD= 13.4) were significantly older than the mental health sample (mean age= 41.3, SD= 13.8) and those who took part in the physical health study were significantly older (mean age= 44.4, SD= 13.4) than non-responders (mean age= 41.0, SD= 13.3).” (pages 12-13, lines 281-9)

4. Lines 288 – I would recommend choosing whether you want to refer to this as a “mental health survey” or “anxiety survey” and being consistent throughout. If you want to refer to it as an “anxiety survey” then the sample should also be described as such instead of the “mental health sample.”

We agree. We have now used mental health survey/sample throughout after describing in the methods section (page 6, line 124-5) that the survey focussed on anxiety as a common exemplar of mental health.

5. Lines 288-307 – it is unclear if these two samples were sent out different surveys for the present study, or if the information about medication and service use was pulled from the parent studies from which the participants in this study were recruited. If they were sent out distinct surveys for this study, then that should be clarified in the methods section. If this data was obtained from the parent studies from which these individuals were recruited, that should be clarified in lines 288-307.

Lines 288-307 (as you reference) are no longer included in the manuscript as part of our response to the next point that you raise. We have however added further information regarding the collection of this data under the 'measures' subheading of the methods section and we hope that this adds clarity (see below).

“Both samples completed the same survey about adjustments to services. Both samples also provided data on prior service use and experiences of being offered adjustments. Other data were also collected as part of the PAT-A and IHOAP studies and will be reported separately (manuscripts in process)”. (page 9, lines 189-90)

- Previous comment that was not adequately addressed in this revision – “While the description of service use is interesting, it is not clear how this relates to the objectives of this study. This information could be summarized in a table rather than taking up valuable space in the manuscript. I would rather see the EFA methods brought back into the manuscript (rather than being supplemental) and this service use section cut since it is not directly linked to the objectives.” I understand that you want to include this to provide context about the individuals who participated in the study, but this could be summarized in a table -- it could probably be easily incorporated into Table 1. Currently, there 4 full pages of the manuscript (pages 12-15) dedicated to describing the sample rather than using that valuable space to address the stated objectives of the study. It certainly is important to describe the sample, but (in my opinion) too much space has been dedicated to this. As a result, findings that are more directly related to your objectives have been placed in the supplemental materials instead of the main manuscript.

We have followed this suggestion and moved information about service use into Table 1. We have then included a very brief text description regarding how the availability (or lack thereof) of adjustments may relate to service use/engagement. We consider this highly relevant to the main aim of this paper. Similarly, we have significantly reduced the text in the results section describing sample characteristics. Details of statistical comparisons can now be found in the supplementary materials with a short summary of the noteworthy findings in the results section of the manuscript.

We have now moved further information regarding EFA (previously in the supplementary materials), to the methods section of the main paper as requested.

7. Previous comment that was not adequately addressed – “Table 3 – Per the methods section, the categories here were: 1) never available, 2) rarely available, 3) available about half the time, 4) available most of the time, 5) available all of the time. It is unclear then why Table 3 presents “rarely or never available” and “never available” as the categories. Does this mean that all of the participants who responded “never available” are also represented again in the “rarely or never available” column?” -- It would make more sense to me to have the columns be “rarely available” and “never available”, that way, if readers want to sum them up, they can and there would not be responses that are included in both columns.

We believe that presenting the ‘important’ and ‘unavailable’ data together more strongly illustrates our message that these adjustments are important yet unavailable to the vast majority of autistic people. That said, we can accept that if this presentation was unclear to you then it may be unclear to other readers. We have therefore changed Tables 2 to report ‘somewhat important’ and ‘very important’ and Table 3 to report ‘rarely available’ and ‘never available’ as you have advised. It is now the case that no responses are repeated in both columns.

Due to the point that we make above, we have still chosen to refer to the combined responses for ‘importance’ and ‘unavailability’ in the written results. In this case, we have made clear that this refers to the addition of the data presented in Tables 2 and 3; see below:

“...those rating each adjustment as important (i.e. ‘somewhat important’ or ‘very important’)” (page 18, lines 326-7).

“For over half of respondents all but three adjustments were endorsed as ‘unavailable’ (i.e. rarely available’ or ‘never available’)” (page 20, lines 362-3).

Discussion

8. Prior comments were adequately addressed.

Reviewer: 1

I thank the authors for their very thorough and thoughtful response to the first review. The revision reads well and the study’s motivation and implications have been adequately clarified. This study is an important contribution to the literature, highlighting unmet needs among autistic adults in both mental and physical healthcare and identifying healthcare adjustments of high priority. I just have very minor suggested edits.

Thank you for your helpful comments and suggestions.

9. Abstract: The three adjustment areas are referred to as “categories” in the results but “factors” in the conclusions. Same terminology should be used in both places for clarity.

Thank you for highlighting this. We have amended the abstract so that the word ‘factors’ is now used consistently throughout.

10. Methods: The authors mention using t-tests and ANOVA to compare across groups. ANOVA is typically used to compare means across several groups. The authors may have meant to say that chi-square tests were used to compare the distribution of categorical variables across groups.

Thank you for highlighting this to us. The use of ANOVA was an error. We have amended the manuscript to correct the error (see below).

“Key demographic characteristics (age, gender and level of autism traits) were compared between groups using independent t-tests to compare means or chi-square tests to compare the distribution of categorical variables...” (page 10, lines 219-21).

11. Please doublecheck the mental health sample sizes reported in the sentence: “In mental health services, those rating each adjustment as important ranged from 46% (n=240) (access to

online appointments) to 98% (n= 525) (clinician who understands autism).” They differ slightly from Table 2.

Thank you for bringing this to attention. The results should read 98% (n= 524) as they show in Table 2. We have amended this in the manuscript (page 18, line 327)

12. As the authors point out, the paired analysis among participants who took both surveys does remove confounding by some measured and unmeasured factors. However, I am having difficulty following the discussion’s point about power (which is related more to statistical efficiency) since the authors did not demonstrate that the two sample design (which is more than double the size of the paired design) was underpowered to detect a significant differences in adjustment importance by service-level. The loss in sample size in the paired analysis is important because the cited Wacholder reference suggests that statistical precision is greater in a paired design compared to a two sample design when the number of observations is the same. While the paired design may actually reduce power in the present study, the tradeoff is the ability to internally control for potential confounding factors and thus obtain a more accurate estimate of the differences between the service-levels. The authors may want to consider reframing the strengths of the paired design not in terms of power (random error) but in terms of bias reduction.

Thank you for raising this point, we have changed the reporting in the manuscript in line with your suggestion (see below). We have removed the Wacholder reference as we do not feel that this point now requires reference to other work.

“However, the use of paired comparisons gives confidence that any identified differences are a true reflection of differences between mental and physical healthcare by virtue of eliminating the potential for sources of between-person variance.” (page 24, lines 480-81)

VERSION 3 – REVIEW

REVIEWER	Jennifer Ames Kaiser Permanente Northern California, United States
REVIEW RETURNED	12-Jan-2021

GENERAL COMMENTS	Thank you for responding to the reviewer comments. The revision is clear, the methods and results are sufficiently detailed, and the discussion offers thoughtful interpretation of the results. Thank you for your hard work on this important study!
--

REVIEWER	Brittany Hand The Ohio State University, USA
REVIEW RETURNED	28-Dec-2020

GENERAL COMMENTS	All previous comments/requested edits have been adequately addressed. Thank you again for conducting this important work and for the opportunity to review.
---